# Task-GAN: Improving Generative Adversarial Network for Image Restoration

## Abstract

Deep Learning (DL) algorithms based on Generative Adversarial Network (GAN) have demonstrated great potentials in computer vision tasks such as image restoration. Despite the rapid development of image restoration algorithms using DL and GANs, image restoration for specific scenarios, such as medical image enhancement and super-resolved identity recognition, are still facing challenges. How to ensure visually realistic restoration while avoiding hallucination or mode-collapse? How to make sure the visually plausible results do not contain hallucinated features jeopardizing downstream tasks such as pathology identification and subject identification? Here we propose to resolve these challenges by coupling the GAN based image restoration framework with another task-specific network. With medical imaging restoration as an example, the proposed model conducts additional pathology recognition/classification task to ensure the preservation of detailed structures that are important to this task. Validated on multiple medical datasets, we demonstrate the proposed method leads to improved deep learning based image restoration while preserving the detailed structure and diagnostic features. Additionally, the trained task network show potentials to achieve super-human level performance in identifying pathology and diagnosis. Further validation on super-resolved face identity recognition tasks also show that the proposed method can be generalized for diverse image restoration tasks.

## 1 Introduction

Image restoration is an essential computer vision task and a widely applied technique. Recently there are increasing interests and significant progresses in this area enabling more realistic image super-resolution Dong et al. (2014); Ledig et al. (2016); Lim et al. (2017); Bulat et al. (2018); Zhao et al. (2018); Yuan12 et al. (2018), in-painting Xie et al. (2012); Yeh et al. (2017); Yang et al. (2017); Ulyanov et al. (2017) and denoising Xie et al. (2012); Zhang et al. (2017b;a). With the development of image restoration technologies, various applications can be applied in different verticals to reach the unfulfilled needs.

Among all the image restoration applications, restoration in medical imaging is one of the most challenging tasks. Image restoration in medical imaging is important and attractive, since it enables imaging in more desirable conditions, e.g. imaging with faster protocols Pruessmann et al. (1999), cheaper devices and lower radiation Naidich et al. (1990), etc. However, medical image restoration requires a tougher evaluation than restoring natural images. It does not only require sharper and visually realistic restoration, but also requires accurate image completion without altering any pathological features or affecting any diagnostic qualities/properties. Therefore medical image restoration can be a benchmark task for related image restoration techniques.

Within this decade, image restoration technique has been rapidly growing by incorporating various prior information into solving the ill-posed inverse imaging task. The prior information evolves from using sparse representation assumption Mairal et al. (2008), enforcing low-rank analysis Dong et al. (2013) to more recently using deep learning based priors Wang et al. (2015) or models Zhang & Zuo (2017). However there are still several challenges and limitations for existing algorithms:

1) Pixel-wise losses for deep learning do not consider non-local structural information which leads to blurred and not visually plausible restoration Ledig et al. (2016).

2) Generative Adversarial Network (GAN) Goodfellow et al. (2014) based methods significantly improve the results to generate visually realistic restoration Ledig et al. (2016). However GANs ensure the consistency to a learned distribution but do not necessarily guarantee the visually plausible solution exactly matches the corresponding ground truth.

3) It is still possible that hallucination or mode-collapses may happen while minimizing loss function designed in improved GAN frameworks Goodfellow (2016), Arjovsky et al. (2017).

4) The discriminator network regularizes on general image distribution and visual quality , but it does not consider what are the key characteristic features such as pathologies, contrasts and image identification that the model needs to preserve for restoring an image.

These challenges are critical for vertical applications such as medical imaging and surveillance where not only the visual property but also the fidelity of the recovered details matters for key purposes of pathology or recognizing faces.

To solve the problems and challenges for realistic and accurate image restoration, we propose the task-GAN which extends GAN based image restoration framework and includes 3 networks: a Generator, a Discriminator and a Task-specific Network. The new task-specific network predicts the pathology recognition or face identity from both the ground truth images and the restored images. It helps to regularize the training of generator and complement the adversarial loss of GAN to ensure the output images better approximate the ground truth images. Task-GAN both achieves realistic visual quality and preserves the important task-specific features/properties, which are related to the end goal for medical imaging restoration and super-resolution face restoration.

The contribution of this work are:

- We propose a Task Generative Adversarial Network framework (Task-GAN) to ensure both visually plausible and more accurate (medical/face) image restoration.

- A Task Network and a task-driven loss are introduced to ensure the preservation of visual details important to the downstream tasks, and more importantly it regularizes the image restoration to be more accurate both quantitatively and qualitatively.

- The method is validated on two in-vivo clinical medical imaging datasets across different modalities, including Magnetic Resonance Imaging (MRI) and Positron Emission Tomography (PET). Additionally, the generalization of the proposed method is further evaluated on a super-resolution face restoration dataset.

- Both quantitative and qualitative evaluations were conducted, including rigorous evaluation by human experts (radiologists) to ensure the image restoration quality and preservation of important visual features.

- Results demonstrate the superiority of the proposed method in image restoration and also show the potential of applying the trained task network for super-human level automatic classification/diagnosis.

- Theory behind the method is further discussed. More justification on how the proposed method improves GAN to approximate one-to-one mapping.The way of how the proposed Task-GAN improves the image restoration may lead to better model design for other applications.

## 2 RELATED WORKS

### 2.1 IMAGE RESTORATION

**Image restoration**, such as image super-resolutionLedig et al. (2016), denoising Xie et al. (2012) and in-painting Pathak et al. (2016) etc., has been rapidly developed and widely applied in various applications. For medical imaging application, the restoration task is often an image de-aliasing task Mardani et al. (2017) to reduce artifacts, or denoising task Manjón et al. (2008) to mitigate the reduced SNR due to low radiation energy or low photon counts.

Conventionally, denoising tasks are conducted by using block matching Mahmoudi & Sapiro (2005) and/or sparse coding Dabov et al. (2009) with the assumptions that natural images generally can be

represented using low-rank and sparse signal models, such that improved SNR can be achieved by enforcing the relationship when solving in a block-based approach.

Recently, deep learning further advances the image restoration capability by learning the nonlinear mapping either locally or globally and recover the high quality image information Burger et al. (2012). Deep Neural Network (DNN) models (such as MLP) as well as the convolutional neural network (CNN) models (such as SRCNNDong et al. (2014), DCNN Xu et al. (2014)) show superiority to learn sparse presentation and are much more efficient than optimization based tools in denoising, deblurring and super-resolution applications. Pre-trained network and perceptual loss Johnson et al. (2016) have also been used to improve the cost-function design considering perception based similarities.

For **medical image restoration**, variable methods based on Deep Learning have also been proposed recently. For example, on MRI reconstruction, which restores image from aliasing inputs, deep learning models and GAN based models Mardani et al. (2017) are used to ensure reconstruction quality. Deep learning approaches are shown to outperform conventional sparsity-regularized optimization (Compressed Sensing Lustig et al. (2007) etc.) based methods, generating more accurate and sharper reconstruction. Similar methods have also shown the potential to restore image from low SNR for low-dose Computer Tomography (CT) Kang et al. (2017) or low-dose Positron Emission Tomography (PET) Xiang et al. (2017) images that are acquired with reduced radiation dose. Deep learning methods using multi-scale CNNs show significant improvements to enable low-dose PET (usually at around $25\%$ radiation dosage compared to standard-dose).

## 2.2 Generative Adversarial Network

**Generative Adversarial Networks (GAN)** Goodfellow et al. (2014) have achieved significant improvements in many tasks such as image generation, image translation and image restoration. GAN also plays a game-changer role for image restoration tasks by generating sharper and realistic restoration Ledig et al. (2016) Isola et al. (2016).

The main idea of using GAN for image restoration is to regularize the training with an adversarial loss function. This forces the generated images to follow a learned distribution so that they are indistinguishable from ground truth images. By training the Generator network $G$ and the Discriminator Network $D$ together, $G$ learns the non-linear mapping to restore image quality, while $D$, challenged by $G$, tries to distinguish whether the inputs are from ground truth images or restored images. The adversarial training approach in a way ensures realistic image quality and sharper details.

One of the challenges for applying GAN in image restoration is to reduce hallucination and mode-collapse Goodfellow (2016). Various improvements in better conditioning, cost functions Arjovsky et al. (2017) and model structures have been proposed to more accurately approximate the one-to-one mapping and further improve the robustness of the adversarial training.

**Improving GAN with multiple networks** is proposed in this work. In general, multiple networks are trained to discriminate different information instead of using a single discriminator network to learn a single information. Related but different ideas have been used in learning multiple networks to improve GAN.

For example in Generative Multi-Adversarial Network model Durugkar et al. (2016), multiple networks are introduced to change the role from a formidable adversary to a forgiving teacher and improve the robustness to mode-collapse. Coupled Generative Adversarial Networks model Liu & Tuzel (2016) generates outputs in multiple domains and learns multiple discriminators for better results in several joint distribution learning tasks. Multi-Agent Diverse Generative Adversarial Networks model Ghosh et al. (2017) forces the discriminator to identify multiple generators to improve high quality and diverse generation.

Another idea related to the proposed method is to incorporate additional information in adversarial training. Works on InfoGAN Chen et al. (2016) have shown incorporating information can learn interpretable representations and better generation. Related idea is also applied in method such as Training Using Privileged Information (TUPI) Vapnik & Vashist (2009), which shows significant improvements to SVM as well as Network training. Additional label information provided by added classifier is also proved to be useful in both image synthesisBazrafkan & Corcoran (2018) and semi-supervised generationLi et al. (2017).

## 3 PROPOSED METHOD: TASK-GAN

### 3.1 DESIGNS

Here we propose the Task-GAN that extends the GAN based image restoration framework Isola et al. (2016). The goal of Task-GAN is to predict the image restoration of images $X$ from the corrupted measurements $\tilde{X}$. In addition, we incorporate further information $Y$ in the learning, which is one or a set of properties of $X$ and important to preserve in the image restoration tasks. For example, the property can be the characteristics or identification information for face image restoration or pathology in medical imaging, which are used as examples in this work.

In general, there are three different networks that are optimized in the training.

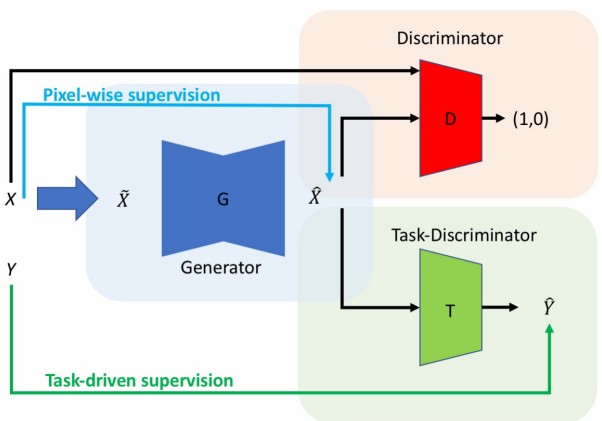

Figure 1: Formulations and flowchart for Task-GAN

**Firstly, a Generator network** $G$, learns the non-linear mapping from inputs $\tilde{X}$ to restoration images $\hat{X}$, which conducts the major image restoration task. This task is supervised with pixel-wise $L_1$ cost function, which has been shown outperform conventional $L_2$ cost function in image restoration tasks Isola et al. (2016).

**Secondly, a Discriminator network** $D$, similar to other adversarial training for GAN, is used to distinguish in the adversarial way to ensure $\hat{X}$ is consistent with the distribution of $X$. A classification task is conducted by $D$ to learn $D(X) = 1$ and $D(\hat{X}) = D(G(X)) = 0$.

**Lastly, a Task network** $T$ is generalized in the multiple image restoration settings. The Task network tries to predict the set of property of $X$, such that it favors $T(X) = Y$ and $T(\hat{X}) = Y$. For a binary case as in pathology recognition example in this work, $Y \in (0, 1)$, representing if there is pathology in the image. Other variant could include classifier for multi-label restoration or segmentation network .

### 3.2 FORMULATION

Figure 1 shows the overall framework of the Task-GAN architecture for image restoration. For a single sample consisting of a image $x$ with its property $y$, we fed the corresponding corrupted image $\tilde{x}$ with the random noise $z$ to the generator which outputs restored image $\hat{x}$. The weights of three networks are optimized based on multiple cost functions across multiple tasks:

1) To approximate the image content in $x$ from generator $G$, we used a pixel-level supervision with $L_1$ loss between $x$ and $\hat{x} = G(\tilde{x})$.

$$L_{pixel} = E_{(x,\tilde{x})\sim p_{data}(x,\tilde{x}), z\sim p_z(z)} \|G(\tilde{x}, z) - x\|_1 \tag{1}$$

2) To stabilize the training process and to challenge the conventional Discriminator Network $D$ which recognizes whether the input is ground truth image $x$ or restored version $\hat{x} = G(\tilde{x})$, we

| Restoration Dataset | Subjects | Images | Task | Task-specific Features Learned |
|---|---|---|---|---|
| 1% low-dose PET | 40 | $\sim 3{,}600$ | Pathology | Diagnostic features for Alzheimer's Disease |
| Multi-contrast MRI | 67 | $\sim 10{,}318$ | Contrast | Non-local features for MR image contrasts |
| Face Images (LFW-a) | 5700 | $\sim 13{,}000$ | Identity | Facial features for identity recognition |

Table 1: Datasets information

used the feature matching adversarial loss Salimans et al. (2016). $f(x)$ denote activations on an intermediate layer of the discriminator.

$$L_{GAN} = \left\| E_{x \sim p_{data}(x)} f(x) - E_{(\tilde{x}) \sim p_{data}(\tilde{x}), z \sim p_z(z)} f(G(\tilde{x}, z)) \right\|_2 \tag{2}$$

3) To teach the Task network $T$ to recognize the property $y$ from image $x$, and more importantly to ensure that the recognizable features are still preserved in $\hat{x}$, we use a regression loss for this task.

$$L_{task} = E_{(x, \tilde{x}, y) \sim p_{data}(x, \tilde{x}, y), z \sim p_z(z)} (T(G(\tilde{x}, z)) - y)^2 + (T(x) - y)^2 \tag{3}$$

### 3.3 FINAL OBJECTIVE FUNCTION FOR TRAINING TASK-GAN

In summary, the optimization task consists of 3 parts: pixel-wise loss using $L_1$ cost, Adversarial loss using feature matching GAN cost and Task loss with regression cost. And the weights from tree networks are optimized to minimize the mixed loss function combining Generator Network $G$, Discriminator Network $D$ and Task-specific Network $T$, for each supervised sample $(x, \tilde{x}, y)$

$$\begin{aligned} L(G, D, T) = &\, L_{pixel}(G, X, \tilde{X}) + \\ &\, \lambda L_{GAN}(G, D, X, \tilde{X}) + \\ &\, \mu L_{task}(T, X, \tilde{X}, Y) \end{aligned} \tag{4}$$

To apply the trained model for image restoration, we pass $\tilde{x}$ through the trained Generator network and the restored $\hat{x}$ is outputted with the model weights to minimize the mixed loss function.

### 3.4 IMPLEMENTATION DETAILS

We adapted our generator and discriminator architectures from Isola et al. (2016). Task-specific network has the Res-Net structure He et al. (2016) and is slightly different in the structures and training schemes. More implementation details can be found in the appendix.

## 4 EXPERIMENTS

### 4.1 DATASETS

The main information of the datasets used is summarized in table 1.

**Ultra-low-dose Amyloid PET datasets**
To evaluate the performance of the proposed method on low-quality medical image restoration, we generated supervised 1% low-dose/standard-dose PET image pairs. Our proposed method was expected to generate high quality synthesized standard-dose PET from 1% low-dose degraded PET images, while preserving pathological features of the Amyloid status (positive or negative) related to the diagnosis of Alzheimer's Disease. 40 subjects were recruited for the study, among which 10 subjects were Amyloid status positive and the other 30 were negative. Datasets were acquired on an integrated PET/MR scanner with time-of-flight capabilities (SIGNA PET/MR, GE healthcare). $330 \pm 30$ MBq of the Amyloid radiotracer (18F-florbetaben) was injected into the subject (as standard-dose) and the PET data was acquired simultaneously 90-110 minutes after injection. The raw list-mode PET data was randomly undersampled by a factor of 100 and then reconstructed as the low-dose PET.

**Multi-contrast MR datasets**

To validate the generality of the proposed method on different modality of medical image restoration, we conducted experiments on magnetic resonance(MR) datasets. In this setting, we tried to synthesize high quality multi-contrast MR neuroimaging from the multi-acquisition input Hagiwara et al. (2017) while preserving the contrast-specific features. The proposed method is aimed to not only generate visually plausible MR images but also preserve the contrast that required for the diagnosis of neurological diseases. 67 cases were included in our datasets. Among them 44 are patients and 23 are healthy controls. Each subjects were scanned with 6 conventional MR sequences(T1w, T1-FLAIR, T2w, T2-FLAIR, STIR and PDw) as the ground truth and Multi-Dynamic Multi-Echoes(MDME) sequence as the input.

**Labeled Faces in the Wild-a (LFW-a)**

To further verify the performance of the proposed method on tasks other than medical image restoration, we adapted it to super-resolution identity-preserving face reconstruction. LFW-a consists of faces captured in an uncontrolled setting with several poses, lightings and expressions. The images were filtered with a Gaussian blur followed 8 times downsampling. Then they were resized to original size by bicubic interpolation as the low-resolution images. We followed the training and test splits as indicated in the LFW development benchmark, which contains a set of image pairs for identity verification task. 1000 images pairs of 500 matched and 500 mismatched pairs for testing verification.

## 4.2 RESULTS

### 4.2.1 QUANTITATIVE RESULTS

Both quantitative and qualitative evaluation were conducted for all the experiments to demonstrate the improvements on image restoration for medical image and natural image applications. Results demonstrate the superiority of the proposed method over the comparable original GAN based solution with improvements in supervised image restoration accuracy as well as the accuracy of the downstream identification tasks of pathology, contrasts and face identity.

**Medical image datasets**

The quantitative results on image quality are shown in table 2. We evaluated the synthesized image quality by peak signal-to-noise ratio (PSNR), structural similarity (SSIM), and root mean square error (RMSE). As shown in the table, for ultra-low-dose PET datasets, the proposed method significantly outperforms the GAN without the task network by 1.50 dB in PSNR, 8.38% in SSIM, and 3.21% in RMSE. Improved restoration is also visible on the Multi-contrast MR datasets. The table shows the results for the most remarkably improved contrast T1 sequence, achieving the improvement of 3.88 dB in PSNR, 10.22% in SSIM, and 36.2% in RMSE. More results on different contrasts can be found in the appendix.

For the ultra-low-dose PET datasets, we also conducted experiments to evaluate the performance of maintaining the pathological features by comparing the error rate of both the Amyloid status network (the task-specific network) and radiologists. We considered the Amyloid reading results on the standard-dose images by two expert radiologists as the ground truth status. They were also asked to read 10 testing datasets based on the synthesized images (volumes) by GAN and task-GAN separately. Amyloid status network's performance was evaluated slice-wise based on the middle 40 slices in each volume by the mean absolute error (MAE). As shown in table 3, radiologists' error rate decreased by 42.8% based on the synthesized images by task-GAN, comparing to the synthesized images by GAN. The Amyloid status network trained ensure consistently super-human level accuracy (no-error) of classification for both standard quality images and low quality images. In addition, the MAE is reduced by 7.4% for the synthesized images using the proposed task-GAN. These results demonstrate that adding the task-specific network can not only improve quality, but also can preserve more task-specific pathological features and potentially enable **super-human level diagnosis** for applications like Alzheimer Disease.

**Natural image dataset**

The extended experiment on super-resolution face restoration can also verify the similar conclusion as in medical image restoration. Apart from the improved image quality metrics of PSNR and SSIM shown in 2, we also evaluate the proposed method by face identity verification. One widely

| Method | Ultra-low-dose PET | | | Multi-contrast MR (T1) | | | LFW-a | | |
|---|---|---|---|---|---|---|---|---|---|
| | PSNR | SSIM | RMSE | PSNR | SSIM | RMSE | PSNR | SSIM | RMSE |
| Input | 22.41 | 0.864 | 0.304 | N/A | N/A | N/A | 22.52 | 0.679 | 0.168 |
| GAN without Task-net | 28.49 | 0.945 | 0.156 | 25.68 | 0.812 | 0.052 | 27.53 | 0.800 | **0.096** |
| Task-GAN | **29.99** | **0.953** | **0.151** | **29.56** | **0.895** | **0.033** | **28.10** | **0.812** | 0.097 |

Table 2: Image quality metrics on the ultra-low-dose Amyloid PET datasets, Multi-contrast MR datasets (T1 sequence), and LFW-a datasets, with the comparison of the GAN without and with the proposed task-GAN feature.

| Classification Performance | standard-dose ground truth images | Synthesized images by GAN | Synthesized images by task-GAN |
|---|---|---|---|
| Human Experts (radiologists) | 0 (ground truth) | 0.35 (7/20) | 0.20 (2/10) |
| Amyloid Status Network (MAE) | 0.139 | 0.135 | 0.125 |
| Amyloid Status Network (Error Rate) | 0 (0/10) | 0 (0/10) | 0 (0/10) |

Table 3: Radiologists and Task-net's performance (Amyloid status) on standard-dose ground truth images, synthesized images by GAN, and synthesized images by task-GAN. Radiologist's pathological decision was based on the whole volume. The table shows the error rate over 2 radiologists' 20 decisions on 10 testing datasets. Amyloid status network was slice-wise, whose accuracy is shown in the table by the mean absolute error (MAE). The volume-wise decision was based on the middle 40 slices in each volume, which is shown in the bracket.

used benchmark Parkhi et al. (2015) for face recognition is extracting features from VGG-Face and computing the Euclidean distance in the embedding space. For the 500 matched and 500 un-matched image pairs in LFW-a, we drew the receiver operating characteristic (ROC) curves in figure 2(a). The area under curve (AUC) of the synthesized images by the proposed method is more than 1% higher than the GAN, which means the additional task-specific network can help to preserve identity related features.

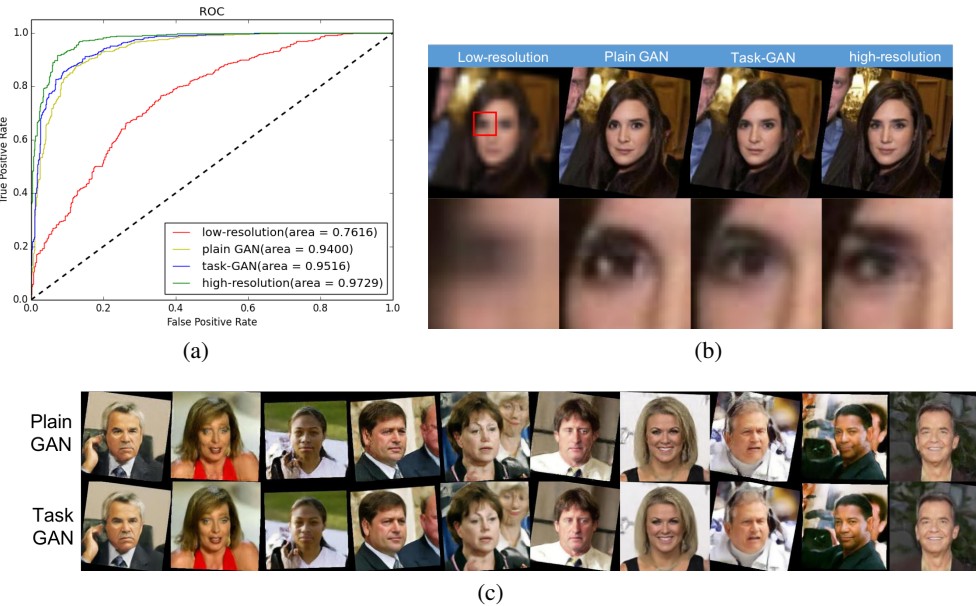

Figure 2: (a) ROC curves for the face identity verification by high-resolution ground truth images, low-resolution input images and synthesized images by GAN and task-GAN. (b) The specific comparative results by GAN and task-GAN. (c) More examples on LFW-a.

### 4.2.2 QUALITATIVE RESULTS

Here we demonstrate the proposed method achieve significantly improved image restoration for multiple datasets. For ultra-low-dose Amyloid PET datasets, a detailed comparison of the restored image is shown in figure 3(a). Radiologists are trained to make diagnosis of Amyloid status positive/negative from the detailed activation pattern on cortex. Figure 3(a)-A and B are Amyloid positive cases while C and D are Amyloid negative. GAN without Task-net blurred out some parts of cortex in A, C and D, and generated some hallucinate uptake in B, while the proposed method kept the original pathological structures better.

For multi-contrast MR datasets, one typical example is shown in 3(b), GAN introduced the artificial features like the grid, while the proposed model achieved better detail-restoration and sharpness. Using task-GAN, the network can get information from contrasts whose quality is better, and store the shared information in the encoded latent space. Also, task-GAN learns the style of different contrasts, and add this additional regularization to the generator to ensure that right features were matched.

For LFW-a datasets, figure 2(b) shows a typical example where the proposed method kept more fine-grained details in faces. Figure 2(c) presents more images comparing the proposed method with GAN's results. As we can see, task-GAN achieved visually better results which have less hallucinate structures and keep more realistic details that related to people's identity.

## 5 DISCUSSION

Results on in-vivo medical imaging datasets demonstrate the superior performance of the proposed algorithm on improved image restoration. The proposed task-GAN achieves this by coupling adversarial training with the training of the task-specific network. Detailed contribution of the task-GAN is explained in the figure 4.

In comparison, the task of the image restoration is to learn a non-linear mapping from low-quality images in the measurement domain to its corresponding high-quality images in a different high-quality domain containing visually realistic images. Shown in figure 4(a), in addition, the recognition of image is a space separation of features/labels along different dimensions that can be orthogonal to the quality dimensions.

In comparison, as is shown in figure 4(b), conventional learning strategy learns the image restoration task by regression, which may fail to generate realistic restoration. The learning is usually based on the minimization of an averaged distance penalty which ensures robustness but lead to unrealistic restoration such as blurring. Additionally, the averaged solution is also likely to be away from the distribution of visually plausible solutions that falls out of the high-quality image space as is shown in the figure.

GAN-based approach on one hand overcomes this by further enforcing an adversarial loss with a Discriminator network which ensures to generate realistic restoration following the distribution of the target high-quality images. As the figure 4(c) shows, the solution is no longer an simple average but pushed into the space of visually realistic high-quality images. However, on the other hand, the discriminator only regularizes the output samples to follow the distribution but ignores the inter-sample relationship. For example, it cannot avoid hallucinations or mode-collapse, where the restored images may be over-similar or undesirably add/remove important visual features. As is shown in the figure, the restored image can have a different label as the ground-truth which fails the purpose of image restoration. We can picture the hallucinations or mode-collapse as a "shrinking" of solution space.

To avoid the possible mode-collapse and ensure a 1-to-1 mapping, various improved GAN models and cost functions have been proposed. For example, Cycle-GAN Zhu et al. (2017) incorporate a cyclic relationship to improve the mapping. However, cyclic relationship does not necessarily lead to exact mappings. The inter-sample relationship as well as the important feature labels can be swapped while still satisfying the cyclic relationship. The illustrating image can be found in the appendix. For example, in the figure, one task label is altered while the cyclic loss is not affected. This may lead to mode-collapse, or specifically a failed image restoration leading to misclassified pathology/normality for medical imaging applications. The consequences of the restoration errors

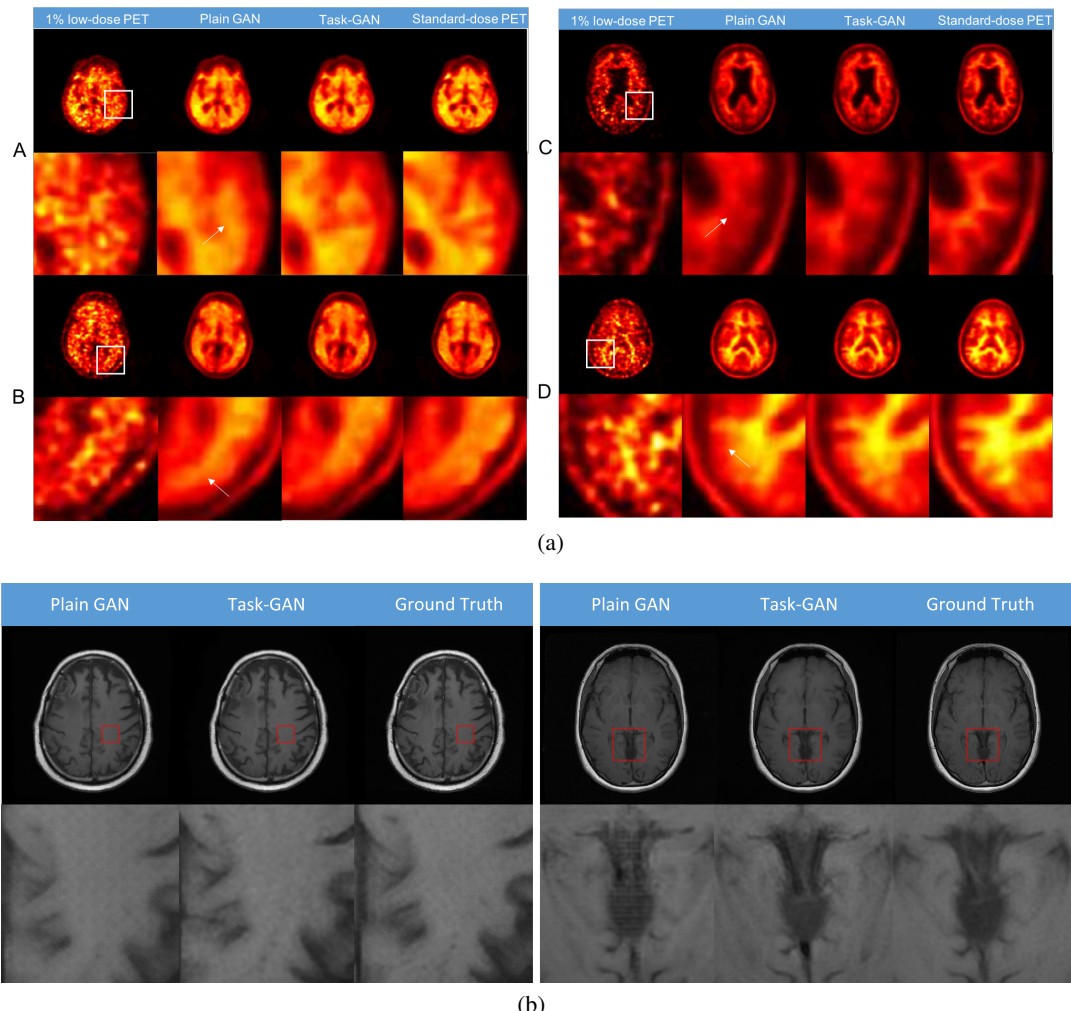

Figure 3: Improved image restoration on (a) Amyloid PET datasets (A and B are Amyloid Positive, C and D are Amyloid Negative) and (b) Multi-contrast MR datasets. Zoomed-in visualization and the error show the pathological important features that are preserved by task-GAN but missing in GAN.

can be huge for medical imaging applications since they can directly lead to mis-diagnosis or over-diagnosis. We can picture the mislabeling or mode-collapse as a "twisting" of solution space. This "twisting" maintains well within visually-plausible space, however severely changes the positioning around the decision boundary of task-label space. More details of the reasoning and visualization will be place in the appendix.

Differently, task-GAN here regularizes both the inter-sample relationship and the sample-label relationship. As is shown in figure 4(d), accurate mapping can be generated with the mixed loss regularization:

1) pixel-level supervision so the restored image is closer to the ground truth,

2) Adversarial loss regularization so that the restored image is within the high-quality space consisting of visually realistic images

3) the task-specific loss that ensure the restored image still preserve the important feature of interests, aka the same labels. In other words, the combination regularization enforce the solution to fall onto the intersection of the manifold preserving pixel-level similarity, distribution consistency and important visual labels. In the view of inter-sample relationship, the task regularization stop the

inter-sample relationship to any visual plausible but destructive "shrinking" or "twisting" around the boundary of task-label space, which ensures more accurate mappings.

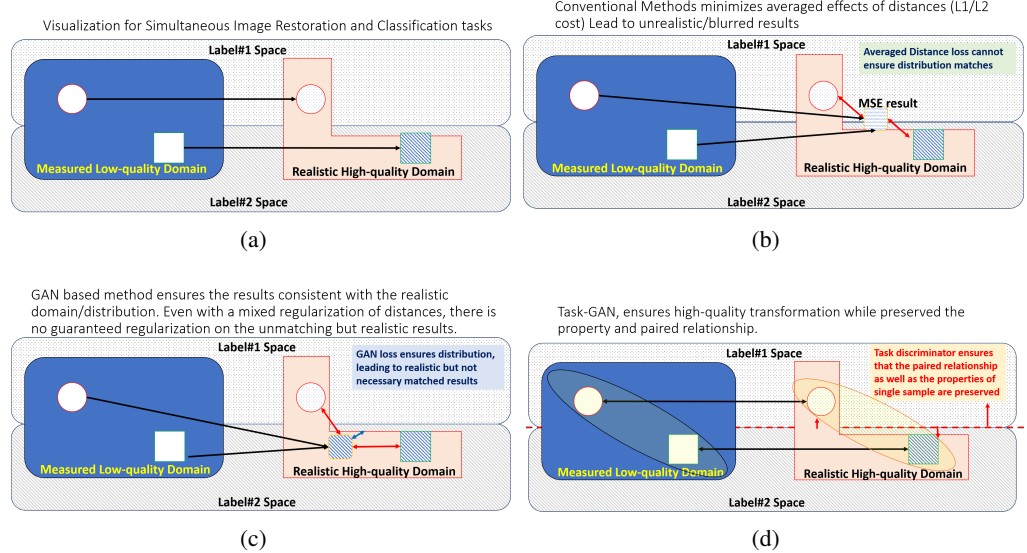

Figure 4: How Task-GAN improves the mapping in image restoration.

# 6 CONCLUSION

In this paper, we proposed an improved design of GAN, Task-GAN, which includes a new task-specific network and corresponding task-specific loss for training GAN based image restoration. Task-GAN is demonstrated to boost the performance of image restoration while preserving important features. Medical imaging applications are used as primary examples, which is one of the most challenging restoration applications since it requires not only realistic restoration, but also high-fidelity as well as accurate classification for subtle diagnostic features. Super-resolution face restoration is used to show the proposed method generalize to natural image applications such as super-resolving face images, where face identity need to be preserved.

The proposed method is demonstrated to achieve superior performance compared with GAN on both image quality metrics and task-specific feature preservation (e.g. pathological features, face identity features, etc.). Based on visual inspection from human experts (clinicians/radiologists), anatomical and diagnostic features are preserved better and fewer artifacts are introduced. The trained task network also shows potentials for super-human level diagnosis tasks.

Task-GAN further extends the regularization of adversarial training. The mixed loss balances between content similarity, distribution consistency and preserving important features for the given tasks. It results in more accurate image restoration with better visual similarity and avoids mode-collapse and hallucinations. Intuitively, task-GAN enforces the solution fall into proper manifold, prevents any alternation ("shrinking" and "twisting") of the restoration from the correct solution space, and preserves both inter-sample relationship and feature-of-interest.

In the future, we will explore further improvements in the design of networks and task formulation. The proposed technique is also valuable to other challenging restoration applications that require realistic restoration and preserving distinguishable details for down-stream tasks.

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

**APPENDIX**

**Network structure**

We adapted the same notation from Isola et al. (2016). $C_k$ represents a Convolution-BatchNorm-ReLU layer with $k$ filters and $CD_k$ denotes a Convolution-BatchNorm-Dropout-ReLU layer with $k$ filters and 50% dropout rate. All Convolution and Deconvolution layers uses $4 \times 4$ kernels and $2 \times 2$ stride. The corresponding layers in the encoder and decoder has skip connections. The architecture of the Generator is:

**encoder:** C64-C128-C256-C512-C512-C512-C512-C512

**decoder:** CD512-CD512-CD512-C512-C512-C256-C128-C64

Discriminator is a image-based classifier with the architecture of C64-C128-C256-C512-C512-C512.

The Task-specific Network is slightly different between different tasks. For the ultra-low-dose PET datasets, the Task Network is a ResNet18 He et al. (2016) pretrained on the ground truth standard-dose images. For the high-resolution face restoration datasets, the Task Network is the pretrained VGG-face model. Parkhi et al. (2015).For the MR contrast-synthesis task, the Task Network is a 6-layer patch-based classifier derived from the discriminator in Isola et al. (2016).

**Training**

All the computation works were done on an Ubuntu server with 4 NVIDIA Tesla V100 GPUs. The proposed network is implemented in TensorFlow and Pytorch. The Adam optimizer is used with $\beta$ chosen at $0.5$ and a learning rate of $2 \times 10^{-3}$. For MR contrast-synthesis, since great variance exists in the target domain(different contrasts), to accelerate the training we adopt a three-step strategy. First the Task Net is trained to learn the features of different labels. Second, only one label(contrast) were trained per epoch, to fasten the convergence of the encoder. Lastly, images with different labels were trained randomly.

**Complementary Figures**

Below is the complementary image for figure 4, illustrating how Cycle-GAN improves the image restoration.

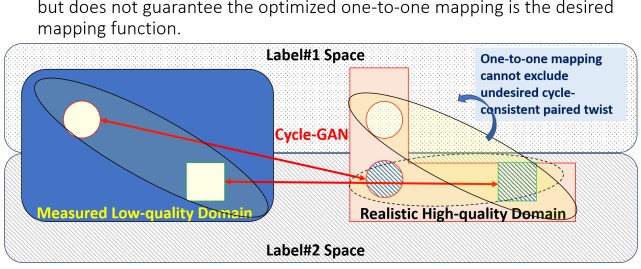

Figure 5: How Cycle-GAN improves the mapping in image restoration.

