# OpenReview forum: "Task-GAN for Improved GAN based Image Restoration"
_ICLR.cc/2019/Conference_

### Official Review · AnonReviewer1 · 2018-10-29
**Interesting applications but limited novelty and poorly selected baseline methods.**

**Rating:** 4
**Confidence:** 5

**Review:**

This paper proposed a new method for image restoration based a task-discriminator in addition to the GAN network. It shows superior performance than the baseline methods without such task-discriminator on medical image restoration and image super-resolution. While the results are better, the idea seems straightforward and has limited novelty. Please see the following comments:

1. Adding an task-discriminator in a GAN network seems straightforward to improve the specific task. And this idea has already used in existing papers, e.g. Cycada.

Hoffman, J., Tzeng, E., Park, T., Zhu, J.Y., Isola, P., Saenko, K., Efros, A.A. and Darrell, T., 2017. Cycada: Cycle-consistent adversarial domain adaptation. ICML, 2018

2. On the application side, the results are not very convincing because the baseline methods were not selected properly. For medical image reconstruction and image super-resolution, the proposed method was not compared with any of the state-of-the-art methods, but only with the same method without a task-discriminator as a baseline. For those tasks, there are many traditional methods and deep nets with different losses. For example, a simple L1/L2 or perceptual loss probably leads to better PSNR than the GAN loss, which is not compared at all. See the attached references.


Ledig, C., Theis, L., Huszár, F., Caballero, J., Cunningham, A., Acosta, A., Aitken, A.P., Tejani, A., Totz, J., Wang, Z. and Shi, W., Photo-Realistic Single Image Super-Resolution Using a Generative Adversarial Network. In CVPR 2017.

Johnson, J., Alahi, A. and Fei-Fei, L., Perceptual losses for real-time style transfer and super-resolution. In ECCV 2016.

Kim, J., Kwon Lee, J. and Mu Lee, K., Accurate image super-resolution using very deep convolutional networks. In CVPR 2016.

3. Some questions about medical image datasets. For the low-dose PET dataset, the input was randomly undersampled by a factor of 100. What is the random pattern? Is it uniform? In addition, why not acquire real low-dose data and show the quality results using the proposed model? For the multi-constast MRI data, how is the input generated and what is the ground-truth?

---

### Official Review · AnonReviewer3 · 2018-11-02
**Novelty is limited and explanation is not clear**

**Rating:** 5
**Confidence:** 4

**Review:**

Authors propose to augment GAN-based image restoration with another task-specific branch such as classification tasks for further improvement.

However, the novelty is limited and not well explained.
1. The idea of adding a task-specific branch has been proposed in Huang et al’s work.
Rui Huang, Shu Zhang, Tianyu Li, Ran He, Beyond Face Rotation: Global and Local Perception GAN for Photorealistic and Identity Preserving Frontal View Synthesis, ICCV 2017.

2. It is not clear why for task-specific loss authors use mse loss instead of cross-entropy loss.
3. It is not clear how much data is used to train the super-resolution model and whether there is overlap between training data for super-resolution task and test data for recognition task.
4. The proposed method is not compared with other super-resolution methods.
5. There are typos with citations. There should be parenthesis around citations.

---

### Official Review · AnonReviewer2 · 2018-11-07
**An interesting paper but incremental technical contribution**

**Rating:** 4
**Confidence:** 5

**Review:**

In this paper, the authors propose a novel method of Task-GAN of image coupling by coupling GAN and a task-specific network, which alleviates  to  avoid hallucination or mode collapse. In general, the paper is addressing an important problem but I still have several concerns as follows:
1. The technical contribution is rather incremental since there exist numerous works on introducing another discriminator to GAN, such as Triple-GAN.

2. Actually, as the authors mentioned, GAN is not an appropriate model for image restoration when  accurate image completion is required. The authors are expected to make comparison with methods not based on GAN framework.

3.  The authors should clarify the details on the Task network since it is non-trivial to model a task.

---

### Meta-Review · Area_Chair1 · 2018-12-06
**Reconstruction GAN with additional classification task, but lacking novelty, evaluation, and references.**

**Confidence:** 4
**Recommendation:** Reject

**Metareview:**

This work presents a reconstruction GAN with an additional classification task in the objective loss function. Evaluations are carried out on medical and non-medical datasets.

Reviewers raise multiple concerns around the following:

- Novelty (all reviewers)
- Inadequate comparison baselines (all reviewers)
- Inadequate citations. (R2 & R3)

Authors have not offered a rebuttal. Recommendation is reject. Work may be more suitable as an application paper for a medical conference or journal.